# Intravaginal Gel for Sustained Delivery of Occidiofungin and Long-Lasting Antifungal Effects

**DOI:** 10.3390/gels9100787

**Published:** 2023-09-29

**Authors:** Andrew Cothrell, Kevin Cao, Rachele Bonasera, Abraham Tenorio, Ravi Orugunty, Leif Smith

**Affiliations:** 1Department of Biology, Texas A&M University, College Station, TX 77843, USA; acothrell@bio.tamu.edu; 2Antimicrobial Division, Sano Chemicals Inc., Bryan, TX 77803, USAatenorio@sanochemicals.com (A.T.); rorugunty@sanochemicals.com (R.O.)

**Keywords:** antifungal, occidiofungin, intravaginal gel, toxicology, RVVC, Franz cells

## Abstract

Fungal infections are caused by opportunistic pathogens that can be life threatening or debilitating. *Candida* spp. are becoming increasingly resistant to current clinically approved antifungal therapeutics. *Candida* infections afflict not only immunosuppressed but also immunocompetent individuals. Recurrent vulvovaginal candidiasis (RVVC) is a disease that afflicts 5–9% of women. Occidiofungin is a novel cyclic peptide that has a broad spectrum of antifungal activity with a novel fungicidal mechanism of action. A gel formulation containing occidiofungin (OCF001) is being developed for use to treat vulvovaginal candidiasis. The formulated gel for intravaginal application used hydroxyethyl cellulose as the primary gelling agent and hydroxypropyl β-cyclodextrin as a solubilizing agent for occidiofungin. Franz cells and LC-MS/MS were used to determine the rate of drug substance diffusion in the gel formulation. The formulation was tested in an ex vivo mouse skin efficacy study, and the safety was tested following repeat intravaginal administration in rabbits. In this study, the gel formulation was shown to reduce the drug substance rate of diffusion across a skin memetic membrane. The study showed that the formulation extends exposure time to inhibitory concentrations of occidiofungin over a 24-h period and supports a single daily application for the treatment of RVVC.

## 1. Introduction

Recurrent vulvovaginal candidiasis (RVVC) is a chronic fungal infection that occurs in an estimated 5–9% of immunocompetent women ages 17 and older [1,2,3,4]. RVVC is a complex disease with an estimated population of over 5 million women in the US alone. One treatment will not fit the medical needs of all people suffering from RVVC. Currently, oral and topical azole formulations are used as a treatment for RVVC. Long-term exposure to these treatment regimens carries the risk of developing additional azole-resistant *Candida* strains, [1,5] thus, potentially putting the patient or others at a greater risk of drug-resistant fungal infections [6,7]. Current therapies on the market do not treat the underlying condition that causes this disease; instead, they attempt to suppress the growth of the fungi (acting as a fungistatic), ultimately relying on the host immune system to clear the underlying infection. Oral fungistatic treatments such as fluconazole and oteseconazole (VIVJOA™) require a long dosage regimen and do have unwanted side effects (nausea, headaches, and unwanted drug interactions) that do not provide an acceptable treatment for all RVVC patients. Increasing the quality of life for those with RVVC remains an unmet medical need.

Occidiofungin (OCF) is a novel antifungal compound that is being developed as a gel product to treat RVVC [8,9,10,11,12,13,14,15,16,17,18]. Structural analyses studies on OCF have shown that OCF is not related to any known therapeutic compound and is a first-in-class antifungal therapeutic [11,13]. OCF is a non-ribosomally synthesized peptide that comprises related structural variants. OCF is a cyclic eight amino acid peptide with a long nonpolar aliphatic chain that is a component of a novel fatty beta amino acid. It is well known that multiple variants can be produced by non-ribosomal peptide synthesis (NRPS) systems. The structural formula of OCF has been determined to vary by the presence of an asparagine or β-hydroxy-asparagine (R1, Figure 1). The structural formula has also been shown to vary by the presence or absence of a xylose sugar (R2, Figure 1). OCF is a fungicidal antifungal agent that has demonstrated a wide spectrum of activity against *Candida* spp. with minimum inhibitory concentrations (MIC) in the low micromolar to nanomolar range [9,10,13,16]. OCF was also shown to be active against caspofungin and fluconazole resistant strains of yeast [16]. OCF has been shown to induce apoptosis in yeast cells, and experiments suggest that OCF interferes with actin cable assembly, leading to the disruption of higher-order actin activities [16].

The use of OCF001 as a new fungicide to treat fungal diseases of the epithelium, such as RVVC, is supported by the mechanism of action of OCF and by the structure of the human vaginal epithelium. The epithelium of the human vagina comprises several cell layers [19,20]. The outermost layer consists of the vaginal stratum corneum (SC), which is a layer of dead flattened cells devoid of cellular activity aside from the synthesis of some new proteins. This cell layer is formed by a process called cornification, which involves the loss of nuclei and intercellular organelles like mitochondria. The vaginal epithelial layer is a highly evolved tissue that is generally resistant to infection and microbial derived toxins. Actin organization in epithelial tissue is unique to that of other tissues and helps form a protective barrier. Further, the outermost layers of the epithelium are undergoing the process of apoptosis, which is the mechanism of cell death induced by the drug substance OCF.

The toxicological and bioactivity studies conducted with OCF to date support the use of a vaginal gel containing OCF to treat RVVC [9,10,15,16,18]. Additional studies are required to determine the diffusion rates of the drug substance and the toxicity of the drug substance in the vaginal cavity. In this paper, we describe gel formulations containing different concentrations of OCF, and these formulations will be tested to determine whether they would be effective and safe for the treatment of a vaginal yeast infection. Data obtained from these experiments informed dosage and drug formulation decisions for OCF001. These observations were used to support an investigational new drug (IND) application with the US Food and Drug Administration. The work conducted in this study will support the development of a first-of-its kind treatment for RVVC.

## 2. Results and Discussions

### 2.1. Drug Substance (OCF)

Previous structural determination studies revealed the major and minor variants of OCF comprising the drug substance [11,13]. The drug substance (OCF) is an amorphous fluffy solid of an off-white color. It is poorly soluble in water (less than 20 µg/mL). Hydroxypropyl β-cyclodextrin is used to enhance the solubility of OCF in the formulated gel drug product (OCF001). It is known that the drug substance is soluble to at least 5 mg of OCF/g of gel based on a high-dose product formulation used in a nonclinical toxicity study [15]. Structures of several known natural variants of OCF (Figure 1) that were identified in the drug substance were analyzed in silico by Derek analysis. The findings from these computational studies showed that the identified variants, presented in Figure 1, were all expected to have the same toxicological properties. Derek analysis showed that the potential of nephrotoxicity in mammals was equivocal due to the presence of a diol moiety on the novel fatty amino acid residue in position 2, and skin sensitization in mammals was also equivocal due to the presence of a phenol on the tyrosine residue in position 4. The results from this analysis support the notion that subtle changes in structures of the isolated variants are equivocal with respect to their potential to cause adverse side effects.

### 2.2. Intravaginal Gel Drug Product (OCF001)

In a previous repeat-dose toxicity study, OCF001 showed no observed toxicity at relatively high concentrations (5 mg of OCF/g of gel) within the intravaginal cavity of mice [15]. OCF was also shown to be a broad-spectrum antifungal and does not have any antibacterial activity [13,16]. Therefore, the activity of OCF001 is selective for yeast and does not have any activity against the vaginal epithelium or the bacteria residing in the vaginal tissue. The proposed treatment of RVVC using OCF001 gel will be selective as a fungicidal, and it is believed that it will aid in restoring a healthy vaginal flora.

The drug product is an aqueous gel formulated using excipients commonly utilized in gel preparations and are generally regarded as safe for use in manufacturing pharmaceutical products. The drug substance (OCF) is not soluble at the desired drug concentration without the use of hydroxypropyl β-cyclodextrin. Propylene glycol and hydroxyethyl cellulose (Natrosol 250 HHX) are common ingredients in gel products and provide the required viscosity needed for an intravaginal gel [21,22,23,24]. Citric acid monohydrate and sodium citrate dihydrate provide buffering to maintain the pH of the drug product suitable for the vaginal cavity (pH of ~4.2) and are within the ranges of pH observed in other intravaginal drug products [25]. Sorbic acid was added as a preservative for the gel product. Sterile deionized water with TDS ≤ 10 ppm filtered by reverse osmosis and a deionization cartridge (or equivalent quality water) was used for production of OCF001 gel formulation. The ingredients used in the formulations are provided in Table 1. Additional tests were performed to ensure that the preparation of OCF001 met the standards for potency and uniformity of an active ingredient OCF. The drug substance concentration was tested by ultra performance liquid chromatography (UPLC) to determine that the drug substance was uniformly dispersed in the gel product (Table 2). Multiple samples were isolated from the gel preparation and tested by RP-HPLC for the drug substance concentration to ensure a uniform dispersion of OCF in the formulated drug product. In these experiments, the OCF concentration was measured within 20% of the expected concentration.

The product excipients were tested to determine whether they interfere with the antifungal activity of OCF. These studies showed that the excipients used in the drug product formulation did not interfere with the activity of the drug substance. The drug substance with the formulated excipients and the drug substance in an excipient-free solution were tested for antifungal activity against *Candida albicans* SC5314 (Figure 2). In Figure 2A, the inhibitory activity of four distinct lots of OCF were tested. The minimum inhibitory concentration (MIC) for all four lots of purified OCF was 0.5 µg/mL. In Figure 2B, the inhibitory activity of OCF was tested in the presence of formulation excipients used to manufacture the drug product (0.150 mg of OCF/g gel). Further, the drug product formulation with a ten-fold increase in the amounts of the hydroxypropyl β-cyclodextrin was also tested to determine whether the solubilizing agent interferes with the OCF inhibitory activity. In both cases, the inhibitory activity of OCF in these conditions remained unchanged at 0.5 µg/mL. In Figure 2C, the zone of inhibition was measured using three samples (Top, Middle, Bottom layer) from a drug product preparation (0.15 mg of OCF/g of gel) that were spotted with 8 µL of gel in the middle of a bioassay plate. All three samples had an equivalent zone of inhibition, supporting the findings that the gel formulation is active and that the drug substance is uniformly distributed in the gel phase. These studies demonstrated that (i) there were no incompatibility issues between OCF and the excipients used in the formulation and (ii) the final gel formulation shows homogeneity with regards to antifungal activity.

### 2.3. OCF Diffusion Rates Determined Using a Franz Cell Apparatus

Franz cell diffusion studies were conducted to evaluate the dosage levels for the intravaginal gel and the influence that diffusion would have on the potency of the gel product over time. Franz cells are designed to measure drug diffusion across a permeable hydrophilic membrane aimed to mimic skin [26]. The data acquired are useful for formulation and justification of drug substance concentrations used in product formulations [26]. A dosage study was conducted to ensure that drug concentrations were maintained above the minimum inhibitory concentration (MIC) levels for a 24 h duration post application. The volume of the vaginal cavity can vary between 4 and 25 cm^3^ depending on age and size [27]. To ensure adequate antifungal potency of the drug product for a single-dose application per day, the concentration of the drug substance is formulated to maintain an OCF concentration above the reported minimum inhibitory concentrations for yeast for 24 h [9,16]. The drug substance concentration is taking into account the maximum volumetric dilution following administration in the vaginal cavity, i.e., a fivefold dilution (5 cm^3^ of gel product into 25 cm^3^ vaginal cavity).

Dosage studies involved placing known quantities of the drug product in a donating Franz cell chamber and measuring the concentration of OCF that crosses an artificial memetic membrane over a 24 h time period. The MIC of OCF against *Candida* species are well characterized to be between 0.5 and 8.0 µg/mL [9,16]. The amount of drug substance remaining in the Franz donor cell at 24 h would need to be 24 µg/mL, taking into consideration a threefold higher dose than the reported high MIC value of 8 µg/mL against yeast and the maximum drug product dilution of fivefold following administration to the vaginal cavity. In this study, two concentrations, 0.150 mg and 0.300 mg of OCF/g gel, were tested to determine whether these concentrations of the drug substance will meet the dosage requirements. All studies were conducted using a Nylaflow membrane; the donor chamber contained 1 mL of drug product, while the receiving chamber contained 5 mL of sodium citrate buffer solution.

The gel drug product was left to diffuse over the course of 24 h into the receiving chamber at a temperature of 37 °C; samples were taken at 0, 0.25, 0.5, 1, 2, 4, 8, 16, and 24 h for LC-MS/MS analyses. After 24 h, the concentration of the receiving chambers was 3.28 and 3.83 µg/mL for the 0.150 mg and 0.300 mg of OCF/g gel samples added to the donor chamber, respectively (Figure 3 and Table 3 and Table 4). Each receiving chamber had 5 mL of buffer solution, giving a total of 16.4 and 19.15 µg of OCF in the receiving chamber for the 0.150 mg and 0.300 mg of OCF/g gel formulations. The equilibrium concentration of the receiving chambers would be 25 µg/mL and 50 µg/mL for 1 mL of the OCF001 gel product containing 0.150 mg and 0.300 mg of OCF/g gel in the Franz donor cell. Given that the concentration detected in each receiving chamber was 87–93% below the expected equilibrium concentrations, it can be inferred that equilibrium was not reached in these assays. The gel-free preparation of OCF also did not reach equilibrium by 24 h. After 24 h, the concentration of the receiving chambers was 9.05 and 13.99 µg/mL for the 0.150 mg and 0.300 mg of OCF/g gel samples added to the donor chamber, respectively (Figure 3 and Table 3 and Table 4). Given the concentration detected in each receiving chamber and the total amount of OCF that diffused across the membrane in the gel free formulation, the chambers were 64–72% below the expected equilibrium concentrations. The data showed that the gel formulation does reduce the rate of diffusion, but also demonstrated that OCF does not rapidly diffuse across a permeable hydrophilic membrane aimed to mimic skin. These data support final drug substance concentrations used in the gel formulations and predict an effective dose of the gel product. The 0.150 mg and 0.300 mg of OCF/g of gel formulations maintained an effective concentration of active ingredient well above the inhibitory concentration of yeast for at least 24 h.

### 2.4. Ex Vivo Efficacy Study Using Excised Mouse Skin

As shown in the above experiments, Franz cells are designed to measure drug diffusion across a chosen membrane. These studies are useful for the determination of drug substance concentration in final drug product formulation [26]. An additional application of this apparatus is to help test and conduct an ex vivo assay on OCF001’s ability to treat a yeast infection considering diffusion. Time-kill studies are traditionally performed to determine how much application time is required to see an effect against the target species. Previous time-kill studies conducted on OCF indicate that a log decrease in cell density after two hours was observed at 1× MIC and 2× MIC [9,17]. This rapid cell density drop is indicative of binding to a critical cellular component essential for fungal survival. Current data suggest actin binding or vesicle trafficking is a critical component for drug substance OCF activity [16,28,29].

Bioassays using the formulated drug substance were performed to better understand the duration and concentration of drug exposure needed for effectively reducing yeast [9,15]. OCF001 drug product at 0.150 mg of OCF/g of gel was tested on excised skin (9 mm diameter) in a Franz diffusion cell. The gel was placed on the surface of infected skin (placebo gel (*n* = 9) and the OCF001 (*n* = 9)). The treatment time was for four hours before the skin was removed and prepped for colony counts. There was a statistically significant log reduction (*p* < 0.001) in *C. albicans* SC5314 burden on skin following a single four-hour treatment compared to no drug treatment control (Table 5). These data demonstrate that the OCF001 product is active at the desired drug substance concentration used in its formulation (0.150 mg of OCF/g of Gel). The data are comparable to the rates of killing observed in liquid media following a short exposure to OCF [9]. Given a 1-log reduction in fungal load with only four hours of exposure demonstrates that the gel product is effective at treating yeast infections.

### 2.5. In Vivo Pathology of Repeat-Dosed Rabbits

In a good laboratory practice (GLP) repeat-dose study in rabbits, pathology was assessed for mortality, gross pathology, and histopathology. For both concentrations of 0.15 mg of OCF and 1.5 mg of OCF/g of gel, no mortality or gross pathological abnormalities were observed in either treatment group. Further, histological examination for all tissues examined was within normal limits. However, microscopic analyses of histopathology showed minimal to marked diffuse inflammatory changes in the vaginal epithelium and underlying mucosal region (Table 6). These inflammatory changes were observed to be infiltration of inflammatory cells that comprise lymphocytes, macrophages, and occasionally neutrophils. These changes were not observed in the recovery groups, which indicates that the changes were considered reversible adverse effects. The lack of any serious events in the 14-day repeat dose experiments, with only minor inflammation observed, demonstrated that the OCF001 gel product formulation is safe.

## 3. Conclusions

Two formulations of OCF001 with 0.150 or 0.300 mg of OCF/g of gel were evaluated for the drug substance to cross a semipermeable hydrophilic membrane that mimics absorption across human skin. The data showed that OCF in the gel formulation does not reach diffusion equilibrium by 24 h. These analyses showed that a drug product formulation of 0.15 mg of OCF/g of gel could maintain a drug substance concentration above inhibitory concentrations for more than 24 h. Further, the ex vivo study showed a log reduction in fungal load following a four-hour exposure. A GLP repeat-dose toxicity study revealed that OCF at a 10× higher concentration of the expected effective dose was safe. Thus, a single daily treatment with OCF001 at 0.15 mg of OCF/g of gel is predicted to be efficacious in the treatment of a human vaginal yeast infection. An effective gel formulation administered intravaginally for the treatment of RVVC would reduce the off-target effects observed with orally administered antifungal agents [30,31,32,33].

## 4. Materials and Methods

### 4.1. Chemicals and Reagents

All reagents were obtained from VWR unless otherwise indicated and were used as received. All solvents used were of HPLC grade or higher and prepared the day of running. The hydroxyethyl cellulose (Natrosol 250 HHX from Ashland) was of a fine particle size and pharmaceutical grade. The internal standard and OCF were prepared by culturing *Burkholderia contaminans* MS14 using processes that were previously reported [11,28].

### 4.2. OCF001 Drug Product Gel Preparation

The percent composition of OCF and excipients are provided in Table 1. The gels were prepared by combining water with citric acid and sodium citrate to make a citrate buffer solution. Citric acid monohydrate and sodium citrate dihydrate were used to produce an acidified buffered gel. The gel is within the normal pH range of the vaginal cavity and will not interfere with the physiological pH of the vaginal cavity. Hydroxypropyl β-cyclodextrin was then added to the citrate buffer, and then the solution was filter sterilized using a 0.2 µm filter. Sterile propylene glycol was heated between 50 and 60 °C, and sorbic acid was then added at a specified amount to the warmed propylene glycol. The sorbic acid was mixed until dissolved. This mixture was allowed to cool, and upon reaching room temperature, hydroxyethyl cellulose was added and mixed until wetted and uniform. OCF was dissolved in the citric acid buffer and then gradually added in small aliquots to the propylene glycol mixture until a smooth and uniform gel was made. The pH of the OCF001 gel product was pH 4.2 +/− 0.2. Propylene glycol is a compound which is GRAS (generally recognized as safe) by the US Food and Drug Administration (21CFR182; Sec. 184.1666). Hydroxypropyl β-cyclodextrin (Cavasol^®^ W7 HP Pharma) is pharmaceutical grade hydroxypropyl-β-cyclodextrin. Cavasol^®^ W7 HP Pharma is a highly soluble β-cyclodextrin derivative. Hydroxyethyl cellulose is pharmaceutical grade and is a nonionic water-soluble polymer for producing clear viscous gels. Sorbic acid is a compound which is GRAS by the US Food and Drug administration (21CFR182; Sec. 182.3089) and is used as a preservative. The prepared OCF001 gel formulations were then stored at 4 °C.

### 4.3. Franz Cell Diffusion Assays

Franz diffusion cells (PermeGear; Hellertown, PA) were used to determine the rate of diffusion across a mimetic membrane for skin. A six-station stirrer containing six 9 mm clear jacketed Franz cells with flat ground joints having a 5 mL receiving chamber and a 1 mL donor chamber were used in the study. Samples were stirred at 600 rpm, and the temperature was maintained at 37 °C using a circulating water heater. Polymeric membranes are usually used for in vitro release testing (IVRT). According to the FDA SUPAC-SS (May 1997), any “appropriate inert and commercially available synthetic membranes of appropriate size to fit the diffusion cell may be used”. A Nylaflo^™^ 0.45 μm polyamide hydrophilic compatible membrane (Cytiva, formerly Pall Lab, Tokyo, Japan) was used in these experiments. The donor chamber contained 1 mL of the OCF001 drug product, while the receiving chamber contained 5 mL of sodium citrate buffer solution. The gel drug product was left to diffuse over the course of 24 h into the receiving chamber, and samples were taken at 0, 0.25, 0.5, 1, 2, 4, 8, 16, and 24 h. A 24 h period was chosen given that the drug product will be dosed once per day for three to five days. These samples collected from the receiving chamber were spiked with internal standard (N^15^ labeled OCF at 1000 ng/mL) for LCMS analysis of OCF concentration.

### 4.4. LC-MS Parameters

A ThermoFisher^™^ Quantum Access triple quadrupole mass spectrometer was used for the mass spectrometry quantification of OCF in the receiving chambers of the Franz cells. The LC-MS/MS analyses were performed using the intact parent mass and the xylose free product masses for quantification (Table 7). A HESI ion source in positive ionization mode was used for data acquisition. The LC-MS/MS was set to monitor in MRM. OCF was monitored using 1216 m/z in Q1 and its product ion 1084 m/z in Q3. The SIL internal standard (N^15^ labeled OCF) was monitored using 1227 m/z in Q1 and its product ion 1095 m/z in Q3. The scan time for both samples was 1.000 s. The liquid chromatography was performed using an Agilent 1100 system with binary pump. A SinoChrom ODS-BP 3 μm 2.1 mm × 50 mm (Dalian Elite Analytical Instruments Co., Dalian, China) column was used, and the column temperature was set at 40 °C. The HPLC flow rate was 0.5 mL a minute with a starting gradient of 95:5 (water:acetonitrile) with 0.1% formic acid. A linear gradient to 40:60 (water:acetonitrile) for 15 min was used to separate and isolate the OCF drug substance. The column was then allowed to equilibrate for 2 min with an isocratic flow of 95:5 (water:acetonitrile).

### 4.5. Ex Vivo Efficacy Study Using Excised Mouse Skin

Sterile Franz cells were assembled, and 5 mL of spider media was added to the receiving chamber and allowed to warm to 37 °C using a circulating water bath attached to the jacketed vessels. Shaved mice skin was placed in the junction between the donor and receiving chamber and clamped to make a fully assembled cell. The excised skin was used as the semipermeable membrane in the ex vivo assay. *Candida albicans* SC5314 was grown in yeast extract peptone dextrose (YPD) broth for approximately 24 h to saturation; this culture was then diluted to OD_600_ of approximately 0.1 in spider media (1% nutrient broth, 1% mannitol, 0.2% K2HPO4) and incubated at 37 °C while shaking until an OD_600_ of approximately 0.7 was reached. Spider media was used to induce a hyphal growth state. Then, 1 mL of the spider media containing the yeast was added to the donor cell of each Franz cell and allowed to incubate for 16 h to allow the yeast to grow and penetrate the excised skin. After 16 h, the spider media in the donor compartment was removed and replaced with 1 mL of OCF001 gel product at 0.150 mg of OCF/g of gel (treatment group) or placebo gel at 0.0 mg of OCF/g of gel (control group). At this time, active diffusion was mimicked by constantly drawing spider media out of the receiving cell and replacing it with fresh spider media; this was performed for 4 h. After 4 h of drawing and replacing media from the receiving chamber (mimicking metabolism), the skin was removed from each Franz cell and vortexed for 1 min in 10 mL of yeast peptide dextrose (YPD) broth containing 50 µg/mL of chloramphenicol. The broth was serially diluted onto fresh YPD media for colony numeration on YPD agar plates containing 50 µg/mL chloramphenicol. The plates were incubated at 37 °C for 48 h and counted to determine fungal load on the excised skin. The data were collected from three independent experiments performed using triplicate samples.

### 4.6. In Vivo Pathology of Repeat-Dosed Rabbits

A GLP repeat-dose pathology study was conducted using New Zealand white rabbits (6 rabbits per group) that were distributed into 4 main groups and 3 recovery groups (Bioneeds India Private Limited; Bangalore, India). Group 1 consisted of control animals that received 1 mL of saline. Group 2 consisted of control animals that received 1 mL of placebo gel product (0.0 mg of OCF/g of gel), while group 3 consisted of animals that received 1 mL of 0.15 mg of OCF/g of gel product. Group 4 consisted of animals that received a high dose of 1.5 mg of OCF/g of gel. Additional rabbits for Group 1, Group 2, and Group 4 comprised the 2-week recovery groups. Saline control, placebo control, and gel product were administered once per day during week 1 and week 4 for each animal. Animals were observed for clinical signs of toxicity, mortality, and morbidity. On day 29 and 43, all animals from the main and recovery groups, respectively, were euthanized by intravenous administration of sodium thiopentone followed by exsanguination and subjected to detailed gross pathological examination. All the organs were collected and preserved in 10% neutral buffered formalin (NBF). All organs/tissues from the main and recovery groups were processed by the paraffin embedding technique, sectioned at 4–6 µm thickness, stained with hematoxylin and eosin, and subjected to histopathological examination by a licensed pathologist. Tissues examined include adrenal, aorta, brain, caecum, cervix, colon, duodenum, esophagus, heart, ileum with Peyer’s patches, gall bladder, jejunum, kidneys, liver, lungs, mammary gland, mesenteric lymph nodes, mandibular lymph nodes, ovaries, pancreas, rectum, sciatic nerve, skeletal muscle, spinal cord, spleen, sternum with marrow, stomach, thymus, thyroid glands, trachea, urinary bladder, uterus, and vagina.

## Figures and Tables

**Figure 1 gels-09-00787-f001:**
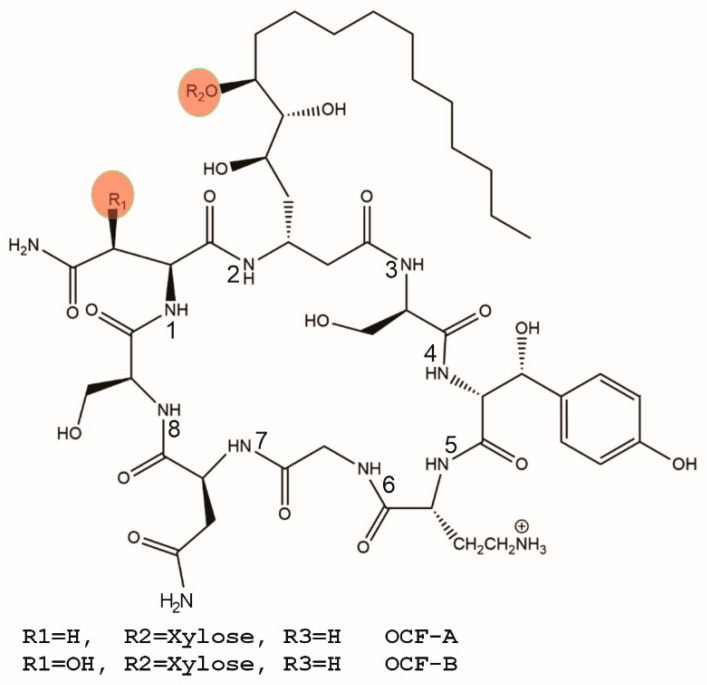
Representative covalent structure of occidiofungin (OCF). Occidiofungin is known to vary in composition at R1, forming asparagine or β-hydroxy asparagine residue. The R2 position is known to vary by presence or absence of a xylose.

**Figure 2 gels-09-00787-f002:**
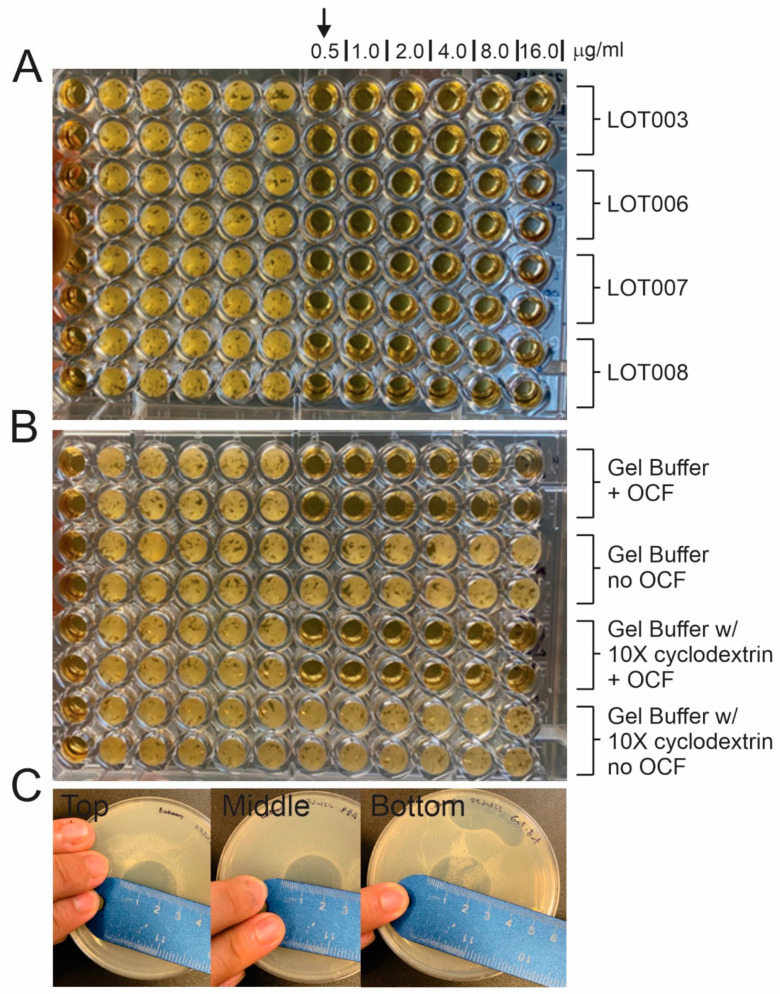
Bioassays with drug substance (OCF) and formulated drug product OCF001 (0.150 mg of OCF/g of gel). (**A**) MICs against *Candida albicans* SC5314 were tested according to a modified CLSI M27-A3 method using OCF from LOTs: 003, 006, 007, and 008). (**B**) The bioactivity of OCF in excipient buffer solution used in drug product formulation and with excipient buffer solution with 10× increase in hydroxypropyl β-cyclodexrin for the 0.15 mg of OCF/g of gel formulation. No OCF negative controls were tested to show that the excipients do not have any antifungal properties. (**C**). Zone of inhibition tests on three samples (Top, Middle, and Bottom) taken from a gel product preparation.

**Figure 3 gels-09-00787-f003:**
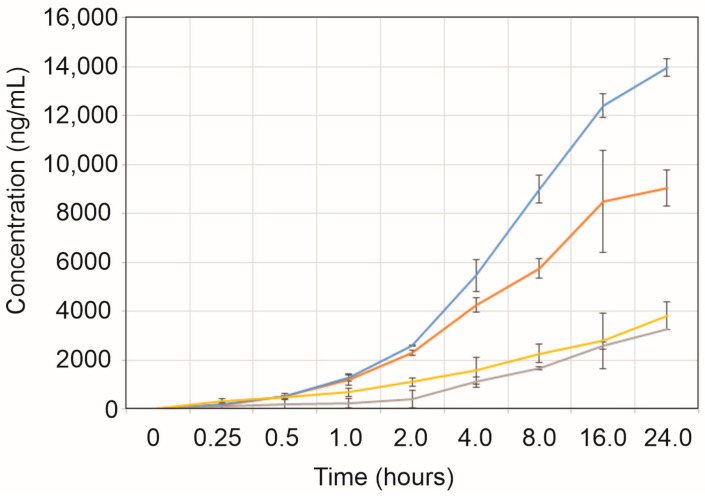
Diffusion of occidiofungin across a permeable hydrophilic membrane. Two drug concentrations and gel/no gel formulations were tested. Grey (0.150 mg of OCF/g of gel), yellow (0.300 mg of OCF/g of gel), orange (0.150 mg of OCF/mL no gel), and blue (0.300 mg of OCF/mL no gel) samples were compared over a 24 h time period.

**Table 1 gels-09-00787-t001:** Formulation of OCF001 intravaginal gel product. The four formulations used in the study are provided.

**Ingredient**	**Composition Percentage (*w*/*w*)**
OCF001-(0.300 mg/g of gel)	
USP purified water	89.510%
Propylene glycol	7.00%
Hydroxypropyl β-cyclodextrin	0.30%
Hydroxyethyl cellulose (Natrosol 250 HHX)	2.00%
Citric acid monohydrate	0.580%
Sodium citrate dihydrate	0.480%
OCF drug substance	0.030%
Sorbic acid	0.100%
**Ingredient**	**Composition Percentage (*w*/*w*)**
OCF001-(0.150 mg/g of gel)	
USP purified water	89.675%
Propylene glycol	7.00%
Hydroxypropyl β-cyclodextrin	0.150%
Hydroxyethyl cellulose (Natrosol 250 HHX)	2.00%
Citric acid monohydrate	0.580%
Sodium citrate dihydrate	0.480%
OCF drug substance	0.015%
Sorbic acid	0.100%
**Ingredient**	**Composition Percentage (*w*/*w*)**
OCF001-(1.5 mg/g of gel)	
USP purified water	88.19%
Propylene glycol	7.00%
Hydroxypropyl β-cyclodextrin	1.5%
Hydroxyethyl cellulose (Natrosol 250 HHX)	2.00%
Citric acid monohydrate	0.580%
Sodium citrate dihydrate	0.480%
OCF drug substance	0.15%
Sorbic acid	0.100%
**Ingredient**	**Composition Percentage (*w*/*w*)**
OCF001-(0.000 mg/g of placebo gel)	
USP purified water	89.690%
Propylene glycol	7.00%
Hydroxypropyl β-cyclodextrin	0.15%
Hydroxyethyl cellulose (Natrosol 250 HHX)	2.00%
Citric acid monohydrate	0.580%
Sodium citrate dihydrate	0.480%
OCF drug substance	0.000%
Sorbic acid	0.100%

**Table 2 gels-09-00787-t002:** Gel preparation and tested by UPLC for drug substance concentration to ensure uniform dispersion of OCF in the formulated drug product.

**OCF001: 0.150 mg of OCF/g Gel**			
Sample Prep Number	1	2	3
UPLC Concentration 1	0.016	0.020	0.019
UPLC Concentration 2	0.017	0.019	0.019
UPLC Concentration 3	0.017	0.020	0.019
Avg.	0.017	0.020	0.019
Sample Std Dev.	0.001	0.001	0.000
Expected conc. of gel (mg/g)	0.150	0.150	0.150
conc of gel (mg/g)	0.150	0.177	0.171
Conc. % error	−0.02	15.24	12.27
Avg of conc. (mg/g)	0.166		
Avg conc. % Accuracy	9.62		
% Accuracy Std Dev.	8.09		
**OCF001: 0.300 mg of OCF/g gel**			
Sample Prep Number	1	2	3
UPLC Concentration 1	0.020	0.020	0.020
UPLC Concentration 2	0.020	0.020	0.021
UPLC Concentration 3	0.020	0.020	0.020
Avg.	0.020	0.020	0.020
Sample Std Dev.	0.0000	0.0000	0.0006
Expected conc. of gel (mg/g)	0.300	0.300	0.300
conc of gel (mg/g)	0.360	0.360	0.366
Conc. % error	16.67	16.66	18.03
Avg of conc. (mg/g)	0.362		
Avg conc. % Accuracy	17.12		
% Accuracy Std Dev.	0.79		

**Table 3 gels-09-00787-t003:** Drug substance (OCF) diffusion characteristics are determined using a Franz diffusion cell.

OCF Diffusion Characteristics				
Time (Hours)	0.150 mg of OCF/g of Gel	0.300 mg of OCF/g of Gel	0.150 mg of OCF/mL no Gel	0.300 mg of OCF/mL no Gel
	Concentration ng/mL			
0	BLOQ	BLOQ	BLOQ	BLOQ
0.25	96	326	166	170
0.5	179	496	537	545
1	226	680	1216	1268
2	394	1105	2312	2581
4	1099	1560	4246	5456
8	1667	2274	5752	9009
16	2607	2797	8483	12,409
24	3279	3829	9051	13,987

BLOQ—below limit of quantification.

**Table 4 gels-09-00787-t004:** Difference between actual diffusion concentration and theoretical equilibrium diffusion concentration.

Final Concentration vs. Equilibrium Concentrations			
Sample	24-h Concentration (µg/mL)	Theoretical Max (µg/mL)	Percent Difference
0.150 mg of OCF/mL no gel	9.0	25	64%
0.150 mg of OCF/g of gel	3.3	25	87%
0.300 mg of OCF/mL no gel	14.0	50	72%
0.300 mg of OCF/g of gel	3.8	50	92%

**Table 5 gels-09-00787-t005:** Four-hour treatment study using OCF001 gel product (0.150 mg of OCF/g of gel) compared to control (0.0 mg of OCF/g of gel).

Ex vivo Trial with OCF001			
	Control Group	Treatment Group	*t*-Test Two Tail
Trial 1 3/23/22	1.30 × 10^6^	3.70 × 10^3^	0.00032
	4.07 × 10^5^	1.57 × 10^4^	
	3.40 × 10^6^	1.33 × 10^3^	
Trial 2 3/30/22	2.35 × 10^6^	1.58 × 10^5^	
	2.12 × 10^6^	1.16 × 10^5^	
	3.80 × 10^6^	5.00 × 10^5^	
Trial 3 4/6//22	2.70 × 10^6^	1.55 × 10^5^	
	4.50 × 10^6^	6.20 × 10^4^	
	2.40 × 10^6^	3.40 × 10^5^	
Average CFUs	2.55 × 10^6^	1.50 × 10^5^	

CFU—colony-forming units.

**Table 6 gels-09-00787-t006:** Incidences of test item-related microscopic lesions in vagina in GLP repeat-dose study in rabbits.

Group		Saline	OCF001 0.0 mg of OCF/g of Gel	OCF001 0.150 mg of OCF/g of Gel	OCF001 1.5 mg of OCF/g of Gel	Saline; Recovery	OCF001 0.0 mg of OCF/g of Gel Recovery	OCF001 1.5 mg of OCF/g of Gel Recovery
Number of animals examined		6	6	6	6	6	6	6
Number of animals with microscopic lesions		-	-	2	6	-	-	-
Vagina	Infiltrate, inflammatory cells	-	-	2	-	-	-	-
	Inflammation, neutrophilic	-	-	-	6	-	-	-

-: No incidence.

**Table 7 gels-09-00787-t007:** LC-MS/MS masses used in the analyses for the quantification of the drug substance.

Occidiofungin Variants: Parent Masses and Product Masses			
	OCF A	OCF B	N^15^ OCF
Parent Mass	1200.73	1216.49	1227.34
Product Mass	1068.7	1084.7	1095.36

## Data Availability

The data presented in this study are available from the corresponding author upon request.

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
