# Peer review of "Intravaginal Gel for Sustained Delivery of Occidiofungin and Long-Lasting Antifungal Effects"

_gels, 2023, doi:10.3390/gels9100787_

Round 1
Reviewer 1 Report
The manuscript “Hydroxyethyl Cellulose Intravaginal Gel for Sustained Delivery of Occidiofungin and Long-Lasting Antifungal Effects” researched hydrogel containing occidiofungin to treat vulvovaginal candidiasis. I have some comments as follows:
1. Author should write keywords
2. I read the manuscript, too much tables presented in the discussion looked like reporting, I think the linear or volume figures should be showed.
2. Although the author descripted the procession of preparing hydrogel, and listed the formulation of OCF001 intravaginal gel, I think it is better to provide more information such as the weight or mole of each ingredient and the size of hydrogel.
3. The author should provide more figures of antibacterial characterization in vitro of hydrogels such as log reduction
Should be improved in disscussion.
Author Response
- Author should write keywords
Response: Included in revised manuscript
- I read the manuscript, too much tables presented in the discussion looked like reporting, I think the linear or volume figures should be showed.
Response: The data in figures and tables has been carefully checked for clarity in revised manuscript
- Although the author descripted the procession of preparing hydrogel, and listed the formulation of OCF001 intravaginal gel, I think it is better to provide more information such as the weight or mole of each ingredient and the size of hydrogel.
Response: The composition (w/w) provides the most direct approach to show gel preparation without the need for calculations
- The author should provide more figures of antibacterial characterization in vitro of hydrogels such as log reduction
Response: The manuscript provides standard MIC assays for characterizing drug substance activity, zones of inhibition for gel preps, and CFU counts for vehicle and gel containing occidiofungin
Reviewer 2 Report
Dear Authors
Many thanks for an article that combines the formation of accurate gels and Diffusion characterization. Results show very interesting possibilities to use the functionalized b-cyclodextrines and hydroxyethylcellulose as the basis for the formation of gels that show control of the delivery of the active principle.
The discussions are very coherent with the results in Franz cells with hydrophilic membrane, as well as in mouse skin and rabbits.
Best regards
Author Response
We appreciate the positive comments and we have made suggested editorial changes to the manuscript.
Reviewer 3 Report
Regarding the review comments on the manuscript “Hydroxyethyl Cellulose Intravaginal Gel for Sustained Delivery of Occidiofungin and Long-Lasting Antifungal Effects” the authors highlighting the increasing resistance of Candida spp. to current antifungal therapies and the need for new treatment options and they highlighting the potential antifungal efficacy of the occidiofungin. They prepared a gel formulation as a new treatment for recurrent vulvovaginal candidiasis (RVVC). They used commonly used and safe excipients in the gel formulation such as hydroxypropyl β-cyclodextrin to solubilize the drug substance at the desired concentration, propylene glycol and hydroxyethylcellulose to provide the required viscosity needed for an intravaginal gel, Citric acid monohydrate and sodium citrate dihydrate provide buffering to maintain the pH of ~ 4.2, and Sorbic acid as a preservative for the gel product.
The manuscript could benefit from the following comments to make it suitable for publication in Gel.
1. The title is not suitable and should be improved. why “Hydroxyethyl Cellulose” in the title?
2. The abstract could benefit from providing more details on how the gel formulation was prepared and analyzed.
3. The specific results and numerical values of the important findings should be included in the abstract.
4. The keywords are missing and should be provided.
5. The first mention of OCF001 was at line 59 without identifying what does it stand for?
6. Table 1 must be improved.
7. hydroxypropyl β-cyclodextrin& propyl β-cyclodextrin, both were in text, please standardize.
8. Identify UPLC&GLP at the first mention.
9. Revise the sample codes in Table 4.
10. Section of “Chemicals and reagents” should be expanded to include more details on the materials used.
11. The experimental section should be supported with relevant references.
Minor editing of English language required
Author Response
We appreciate the positive comments and we have made suggested editorial changes to the manuscript. Please refer to marked copy of the manuscript that show the recommended edits and suggestions were taken into consideration for the resubmission.
Round 2
Reviewer 1 Report
The data and discussion of the manuscript are reasonable.